# Effect of 433 MHz double-slot microwave antennas for double-zone ablation in *ex vivo* swine liver experiment

Xiaofei Jin[1,2]*, Mengwei Jiang[1], Lu Qian[1], Ling Tao[1,3], Yamin Yang[1,2], Lidong Xing[2], Zhiyu Qian[1,2], Weitao Li [ID][1,2]*

1 Department of Biomedical Engineering, College of Automation Engineering, Nanjing University of Aeronautics and Astronautics, Nanjing, China, 2 Key Laboratory of Multi-modal Brain-Computer Precision Drive, Industry and Information Technology Ministry, Nanjing University of Aeronautics and Astronautics, Nanjing, China, 3 Key Laboratory of Brain-Machine Intelligence Technology, Ministry of Education, Nanjing University of Aeronautics and Astronautics, Nanjing, China

* jinxiaofei@nuaa.edu.cn (XL); liweitao@nuaa.edu.cn (WL)

## Abstract

### Purpose

To evaluate the effects of axial length and slot-to-slot distance of double-slot microwave antenna (DSMA) with frequency of 433 MHz on the size and shape of ablation zones created under different input microwave powers.

### Materials and methods

The design of double slot microwave antennas (DSMAs) with axial lengths (70 mm, 30 mm) and slot-to-slot distance (49 mm, 10 mm) were optimized by numerical simulation and *ex vivo* liver experiments. Finite-element method simulations and forty ablations of swine liver were employed to obtain the temperature distributions within liver tissue using DSMAs at the 433 MHz operating frequency in a range of heating powers (20, 30, 40 and 50W) for 600 s. The dependence of the effectiveness of MWA on the axial length and slot-to-slot distance of antenna as well as the input power was further evaluated by analyzing morphologic characteristics of ablated zone.

### Results

Two-zone ablation was achieved by two types of double-slot antennas in our study with frequency of 433 MHz, and the observed shapes of *ex vivo* experimental ablation zones were in good agreement with patterns predicted by simulation models. The ablation zone exhibited a 'gourd' shape after the treatment using the antenna with longer axial length and slot-to-slot distance, while the short antenna caused a guitar-shape ablation in liver tissue after MWA.

### Conclusion

The dedicated design of our DSMAs with a frequency of 433 MHz could enable new ablation shapes with controllable dimensions, which can be applied to the clinical treatment of

**Data availability statement:** All relevant data are within the manuscript and its Supporting information files.

**Funding:** This work was supported in part by the National Major Scientifc Instruments and Equipment Development Project Funded by National Natural Science Foundation of China under Grant 81827803 and 81727804, in part by the National Natural Science Foundation of China under Grant 82151311, in part by the Fundamental Research Funds for the Central Universities under Grant NP2024102, NJ2024016 and NJ2024029, and in part by the Jiangsu Funding Program for Excellent Postdoctoral Talent under Grant 2024ZB661.

**Competing interests:** The authors have declared that no competing interests exist.

MWA for gourd-shaped liver tumors and other long-shaped tumors. Furthermore, research can be conducted on how to design the antenna as flexible and use it for the treatment of pulmonary nodules or varicose veins.

## Introduction

Microwave ablation (MWA) has recently become increasingly attractive in treating malignant tumors, especially for liver cancer [1,2]. Microwaves with the frequencies from 900 MHz to 18 GHz have been widely used in thermal ablation, and currently available MWA modalities for medical use are normally operating at 915 MHz and 2450 MHz [3,4]. However, the relatively small ablation zone using the 2450 MHz operating frequency placed restrictions on large tumor destruction. According to microwave theory, a lower microwave frequency is associated with a more significant increase in temperature, which facilitates the heating of deeper tissues. Our previous study showed that [5], compared with antenna with frequency of 2450 MHz, larger ablation area was indeed obtained by our coaxial slot antenna with frequency of 433 MHz. This finding theoretically underscores the efficacy of microwave ablation antennas operating at 433 MHz for the conformal ablation of large or irregular tumors, in comparison to those functioning at 2450 MHz.

A successful MWA process requires sufficient local cancerous tissue killing that covers the entire tumor area without damaging surrounding tissue. In contrast to those benign masses with round or oval shape, a large amount of cancerous tumor tissues usually has an irregular outline. For those tumor lesions with specific shape and size and in proximity to critical structures, it is critical to ensure complete thermal ablation of the complex tumor volume, while avoiding the damage of nontargeted normal tissue or blood vessels [6–9]. Nevertheless, typical ablation zone achieved by single 2450 MHz antenna is usually ellipsoidal or spherical shape, it is thus impossible to obtain adequate tumor-free ablation margin by just using one antenna. While multiple insertions of a single antenna for several times or multiple antennas operating simultaneously could lead to conformal ablation, increased invasiveness and system complexity are always the vital drawbacks [10]. For instance, studies have reported that, in the 2450 MHz dual-antenna ablation process, the ablation zone can either be too large, resulting in damage of the normal tissues, or too small, leaving tumor residues [11]. It has also been reported that various complications could be caused by the damage of intrahepatic portal pedicles, surrounding large vessels and gallbladder during MWA process for liver cancer [12]. Therefore, there is an emerging demand in developing a practical and effective method for ablating tumors with irregular shape and specific location using one single applicator.

The performance of an antenna, particularly its capability for backward heating along the axial direction, is significantly influenced by its structural design. Slit antennas demonstrate higher radiation efficiency and reduced energy loss compared to monopole and dipole antennas, rendering them ideal for microwave ablation needle applications.

In present study, we successfully designed two coaxial slit antennas based on the 433 MHz microwave frequency, so that MWA can be used to treat tumors of a specific shape, size, and location. The design of our double slot microwave antennas (DSMAs) with axial lengths of 70 mm and 30 mm were optimized by numerical simulation and *ex vivo* liver experiments. Different axial length and slot-to-slot distance of DSMAs were also studied to evaluate the effectiveness of MWA in a range of heating powers. The dedicated structure of our DSMAs with frequency of 433 MHz could create novel ablation shapes, including gourd-shape and guitar-shape ablation zones with controllable dimensions, which owns potential for future application of precise and patient-specific MWA.

## Materials and methods

This article does not contain any studies with human participants or animals performed by any of the authors. All experiments were carried out ex vivo using swine livers and no animal was specifically sacrificed in our study. Fresh swine liver was obtained from the local slaughterhouse. The experimental design, simulation study, *ex vivo* experiments, data collection and analysis were performed by the authors without relevant interest conflicts. All authors contributed to the present work and manuscript preparation.

### Antenna design

In this study, two coaxial slit microwave antennas have been designed, both of which use semi-rigid coaxial cable (SFT-50-1) with a characteristic impedance of 50 Ω. The specifications of the cable include an inner conductor diameter of 0.3 mm, an insulating dielectric diameter of 0.8 mm, and an outer conductor diameter of 1.18 mm. At the top of each antenna, the inner and outer conductors are welded together to form a conductive cap. By removing the rings from the outer conductor of the coaxial cable slots are formed, which are channels for the electromagnetic wave radiation energy to the biological tissues. The theoretical value of the slit width ($l$) is related to the effective wavelength ($\lambda_{eff}$) and the relative dielectric constant ($\varepsilon_r$) and is calculated as follows:

$$l = \frac{1}{2}\frac{\lambda_{eff}}{\left(1+\sqrt{\varepsilon_r}\right)}$$

(1)

The theoretical value of the gap spacing ($D$) is also related to the effective wavelength ($\lambda_{eff}$), which needs to meet the requirement of equal amplitude and same phase feeding, and the general calculation formula is shown below:

$$D = \frac{1}{2}\lambda_{eff}$$

(2)

The dimensions of DSMAs are schematically illustrated in Fig 1. Two structures of DSMAs with the same radius of a 50 Ω semi-rigid coaxial cable (SFT-50-1) were modeled. The dimensions of antenna structure were optimized by the comprehensive consideration of theoretical calculation based on effective wavelength (λeff) and previous experiences [5]. The DSMA with axial length of 70 mm (about 3λeff/4 at 433 MHz) was defined as long antenna, whereas the DSMA with axial length of 30 mm (about λeff/3 at 433 MHz) was defined as short antenna. The slots with the width of 1 mm in each antenna were created by removing annular portions of the outer conductor of the coaxial cable [13]. Different tip-to-slot distance and slot-to-slot distance were investigated in corresponding antenna, respectively, and the specific structural details are illustrated in Fig 1.

The ablation needle features a conical front end and a cylindrical rear end, with a PTFE dielectric sleeve firmly attached to the slit, ensuring a seamless connection to the 304 stainless steel needle tube. Semi-rigid coaxial cable is passed through the needle bar and connected to the RF connector for connection and propagation of the 433 MHz microwave source. The water-cooled section is connected to the inlet and outlet pipes via equal-diameter plastic four-way fittings to form a simple tank structure. Cooling water is injected through a capillary tube and flows through the working part of the needle body and then out to achieve circulating water cooling. This design effectively mitigates the trailing phenomenon and tissue carbonization during the ablation process, thereby optimizing the outcomes of ablation.

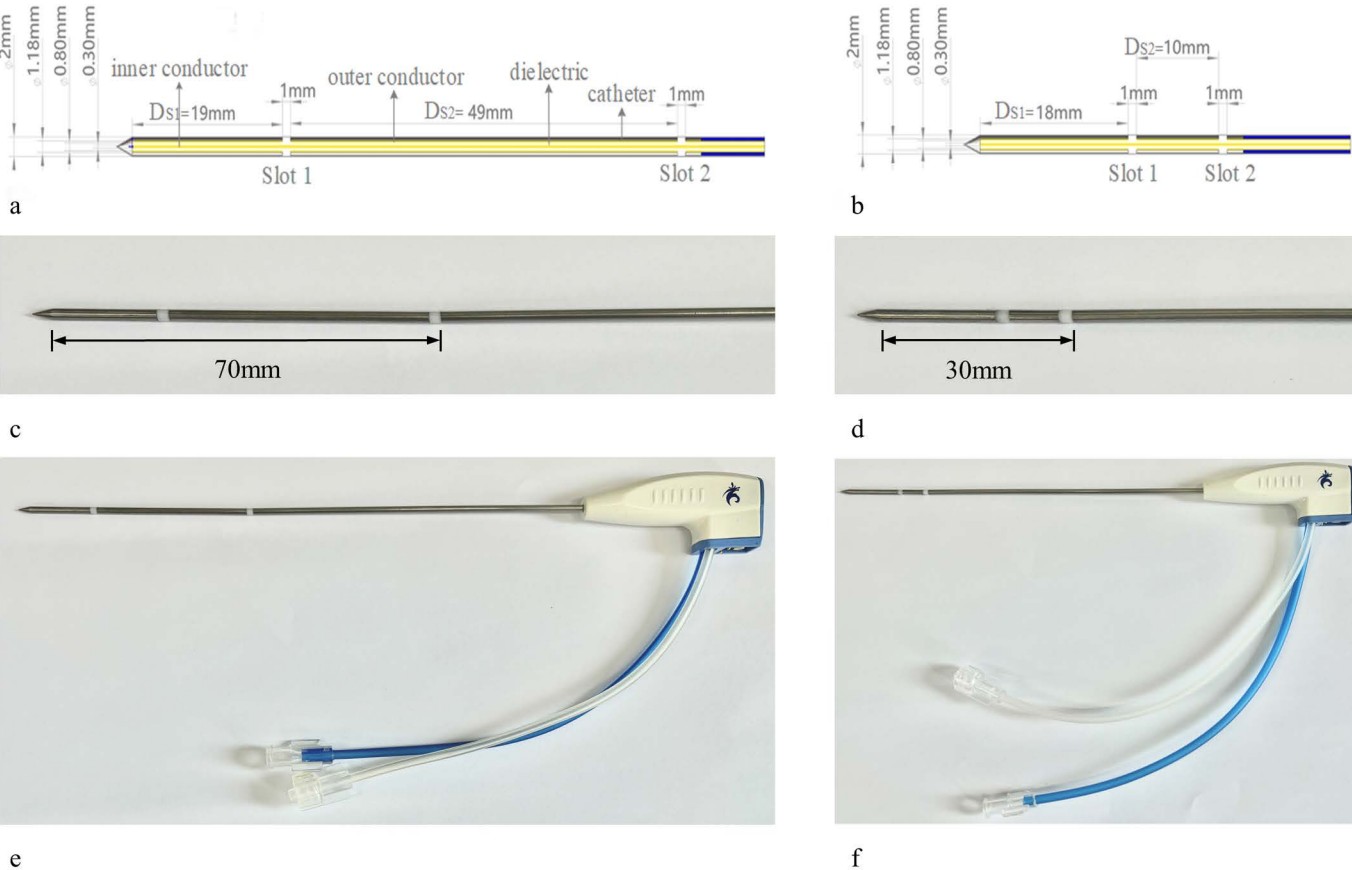

**Fig 1. Schematic illustration and photograph of DSMAs.** Longitudinal side view of (a) long antenna with axial length of 70 mm (b) short antenna with axial length of 30 mm; photograph of (c) the long antenna (d) the short antenna; (e) the DSMA applicator (the long antenna). The tip-to-slot distance (DS1) is 19 mm and slot-to-slot distances (DS2) is 49 mm in the long antenna, respectively. The slot near the tip is defined as slot 1, and the other one is defined as slot 2; (f) the DSMA applicator (the short antenna). The tip-to-slot distance (DS1) is 18 mm and slot-to-slot distances (DS2) is 10 mm.

## Numerical simulation study

The numerical simulation models of swine liver and antennas were set up in COMSOL Multiphysics (COMSOL 5.1, Stockholm, Sweden) using Finite Element Method (FEM). Since the isolated pig liver serves as a more homogeneous medium, we assume that the liver tissue is isotropic. This allows us to simplify the simulation model of the liver tissue, which originally possesses an irregular shape, into a two-dimensional axially symmetric model. In the ablation experiment, the depth of insertion for the 433 MHz microwave ablation needle into the isolated liver tissue will not exceed 14 cm. Consequently, we set the height of the three-dimensional cylinder to 140 mm and its transverse diameter to 50 mm. The depth of antenna insertion into the liver tissue will be twice that of its axial length; thus, for long antennas it will measure 140 mm and for short antennas it will measure 60 mm.

The simulation of the temperature field in the microwave ablation model primarily utilizes a coupled and interdependent electromagnetic wave and bioheat transfer model. In the electromagnetic field distribution of a coaxial microwave antenna, both the electric vector and magnetic vector are perpendicular to the direction of propagation; specifically, the electric field has values only in the radial (r) component, while the magnetic field is present solely

along the azimuth (φ) axis. Therefore, this paper employs a transverse magnetic (TM) wave equation as the model for electromagnetic wave propagation, represented by the following equation:

$$\nabla \times \left( \left( \varepsilon_{\mathrm{r}} - \frac{j\sigma}{\omega\varepsilon_0} \right)^{-1} \nabla \times H_{\varphi} \right) - \mu_{\mathrm{r}} k_0^2 H_{\varphi} = 0$$

(3)

Where, $\varepsilon_0$ indicates the relative permittivity in a vacuum, which has the value of $8.854 \times 10 - 12$ F/m), $\varepsilon_r$, $\sigma$ and $\mu_r$ are the relative permittivity, electrical conductivity and permeability of liver tissue respectively, $k_0$ denotes the free-space wave number, $H_{\varphi}$ is the magnetic field strength.

The Specific Absorption Rate (SAR) reflects the microwave energy absorbed per unit of time by biological tissues and can be expressed as:

$$SAR = \frac{\sigma|\vec{E}|^2}{2\rho}$$

(4)

Where, $\vec{E}$ denotes the induced electric field in the organization, $\rho$ is the liver tissue density.

After biological tissues absorb microwave energy, the energy is subsequently transmitted within the tissue in the form of heat. This transmission primarily occurs through two mechanisms: thermal conduction and convection. In the field of clinical microwave hyperthermia, accurately predicting and controlling the temperature distribution in tissues following energy absorption is essential. This necessitates the development of a model for heat conduction in biological tissues. Among various models describing temperature field variations, Pennes' bioheat equation stands out as the most classical representation, expressed as follows:

$$\rho C \frac{\partial T}{\partial t} = \nabla \cdot (k \nabla T) + \rho_b C_b \omega_b (T_b - T) + Q_{met} + Q_{ext}$$

(5)

where $\rho$, $C$, and $k$ represent the density, specific heat capacity, and thermal conductivity of liver tissue, respectively. In the right-hand term of the equation, $\nabla \cdot (k\nabla T)$ accounts for heat transfer between tissues, while $\rho_b C_b \omega_b (T_b - T)$ denotes the blood flow action term. The variables $\rho_b$, $C_b$, $\omega_b$, $T_b$ correspond to the density, specific heat capacity, perfusion rate, thermal conductivity, and temperature of blood. Additionally, terms $Q_{met}$ and $Q_{ext}$ are defined as representing metabolic heat production within the tissue and as denoting heat load applied by an external source (specifically from microwave electromagnetic fields), respectively. The external heat source term is associated with Specific Absorption Rate (SAR) and can be further expressed as follows:

$$Q_{ext} = \rho SAR = \frac{1}{2}\sigma \left|\vec{E}\right|^2$$

(6)

According to Eq. 6, the temperature field during microwave ablation can be simulated by solving both the planar transverse magnetic field wave equation and the Pennes' bioheat equation.

In numerical simulations, the thermal and dielectric properties of the liver tissue model are critical biophysical parameters that vary with tissue temperature and water content throughout the heat transfer process in biological tissues. This paper presents expressions below that incorporate appropriate corrections based on previous literature:

$$\rho \ (kg/m^3) = \begin{cases} 1060 & 20°C \leq T \leq 100°C \\ 1227.6 & 100°C \leq T \leq 200°C \end{cases} \tag{7}$$

$$C \ (J/(kg \cdot °C)) = \begin{cases} 3600 & 20°C \leq T \leq 103°C \\ 2263600 & 103°C \leq T \leq 104°C \\ 2023.7 & 104°C \leq T \leq 200°C \end{cases} \tag{8}$$

$$k \ (W/(m \cdot °C)) = \begin{cases} 0.512 & 20°C \leq T \leq 100°C \\ 0.204 & 100°C \leq T \leq 200°C \end{cases} \tag{9}$$

$$\varepsilon' = 1.178*(48.391*(1-(1/(1+\exp(0.0764*(82.271-T)))))) +1) \tag{10}$$

$$\sigma(s/m) = 0.861*(1-(1/(1+\exp(0.0697*(85.375-T))))) \tag{11}$$

The insulating medium of the microwave ablation needle body was polytetrafluoroethylene (PTFE), characterized by a dielectric constant of 2.03 and zero conductivity. A boundary temperature of 15 °C was applied to the outer conductor surface of the ablation needle model to simulate the effects of circulating water cooling. The simulation model represented ex vivo hepatic tissue, with both blood perfusion rate and metabolic heat production set to zero; additionally, the initial temperature of the hepatic tissue was established at 15 °C.

### *Ex vivo* swine liver study

Fig 2 illustrates the experimental setup for *ex vivo* ablations in swine liver tissue. The MWA system (Fig 2) is mainly composed of: microwave source (Hangzhou Bada Microwave Technology Co, Ltd), coaxial cable, DSMA, and cooling water equipment (LongerPump Co, Ltd). The DSMA is connected to the microwave source which could provide coherent power to the antenna with the frequency of 433 MHz. Room temperature tap water was circulated through the antenna at a flow rate of 0.22 m/s with the cooling water equipment. In our experiment, healthy swine livers (approximately 400 g) were procured from a local slaughterhouse and stored at 4 °C. The microwave source was set with an input power of 20 W, 30 W, 40 W and 50 W for 10 minutes using the fabricated antennas. Five samples were evaluated per power for each antenna.

The swine livers were allowed to equilibrate with at room temperature (15 °C) for 25 min [14] before *ex vivo* experiment. The DSMAs were inserted into the liver tissue at depth of 14 cm and 6 cm for long and short antennas, respectively. Immediately after each ablation, the DSMA was withdrawn and tissue samples were sliced perpendicular to the length of the antenna and then the area of tissue ablation was photographed accordingly. The maximum longitudinal dimension of the ablation zone along the antenna insertion path was defined as the long-axis diameter (LAD), and the maximum dimension transverse to the antenna was defined as the short-axis diameter (SAD). We measured LAD and SAD of each ablation zone from the sliced specimens, and statistical difference of ablation shape and size using various heating powers were further analyzed. Since the two ablation zones are all close to the ellipsoid shape, the volume (V) of the ablation zone was calculated by using the formula for an ellipsoid [15]: V = (4/3)π(LAD/2)(SAD/2)(SAD/2). The sphericity index (SI) of each ablation was calculated by using the formula from Hines-Peralta et al. [16]: SI = [(4/3)π(LAD/2)(SAD/2)(SAD/2)]/[(4/3)π(LAD/2)³]

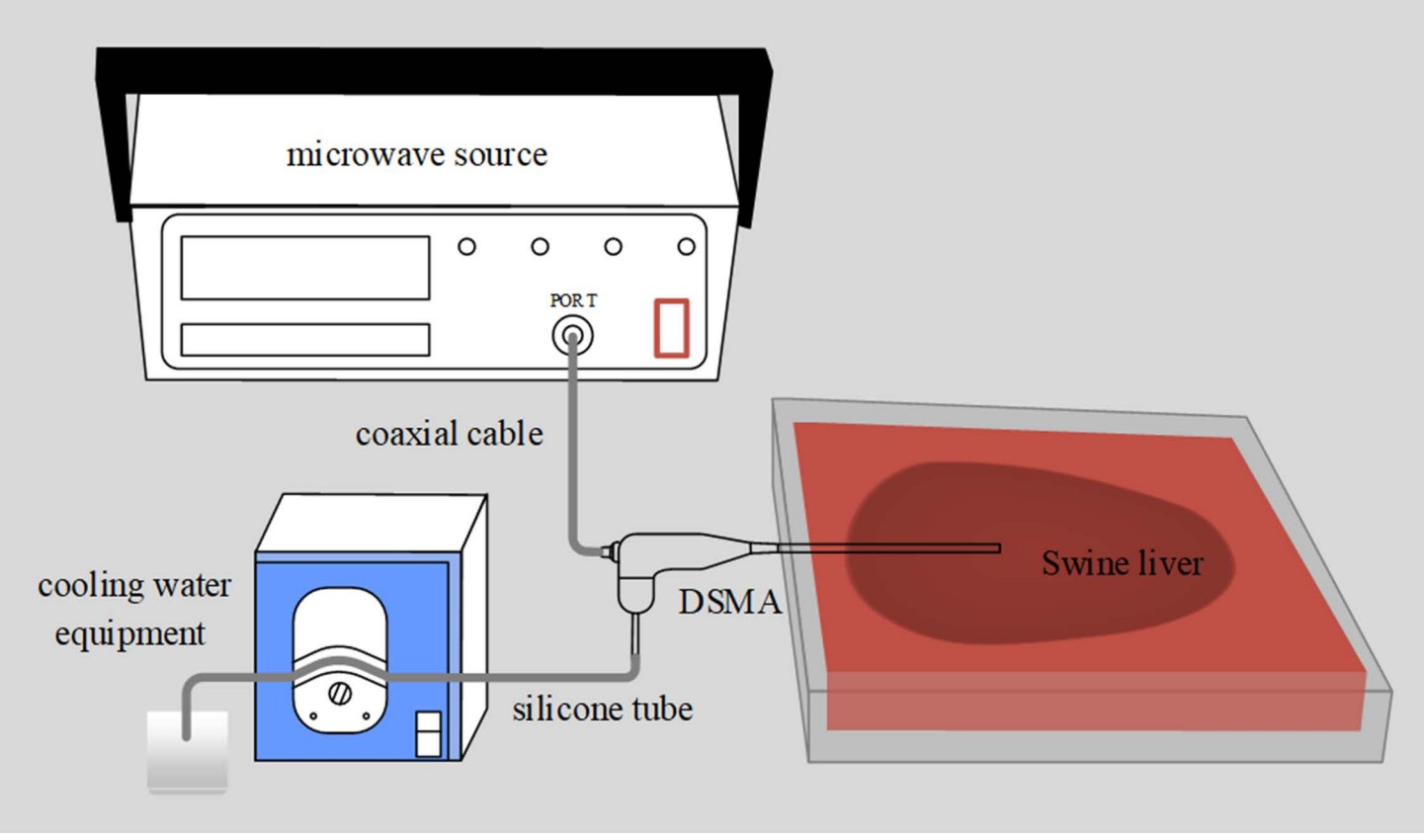

**Fig 2. System for experimental assessment of DSMAs in *ex vivo* swine liver tissue.**

## Statistical analysis

All statistical analyses were performed by using Origin software (OriginPro 8, OriginLab Company, USA) and SPSS software (IBM SPSS Statistics 21.0, New York, USA). Students' t-test was used to evaluate the differences between ablation zone (LAD and SAD) of the long antenna and short antenna, and to identify the differences between ablation results obtained from simulation and *ex vivo* experiment. All hypothesis testing was conducted at an α level of 0.05 by using the Wald test with Kenward and Roger denominator degrees of freedom [17]. A p value of less than 0.05 was considered to indicate a statistically significant difference. Results were reported as means ± SD in five repetitions.

## Results

The temperature distributions within swine liver when using our fabricated DSMAs were obtained by numerical simulation. The temperature profile based on various input microwave powers (20 W, 30 W, 40 W and 50 W) for a heating time of 600 s are shown in Fig 3. The black line in each temperature distribution contour figure represents the margin of 60 °C isotherm, which is considered as the boundary of effective ablation zone [18]. It can be seen that, for both long (upper row in Fig 3) and short antennas (bottom row in Fig 3), the temperature distribution are symmetrical and the over-ablated zone elongates in both radial and longitudinal directions. It is found that DSMA provides two hot spot zones in simulated temperature distributions. In the case of long antenna, the temperature profile exhibited a 'gourd' shape,

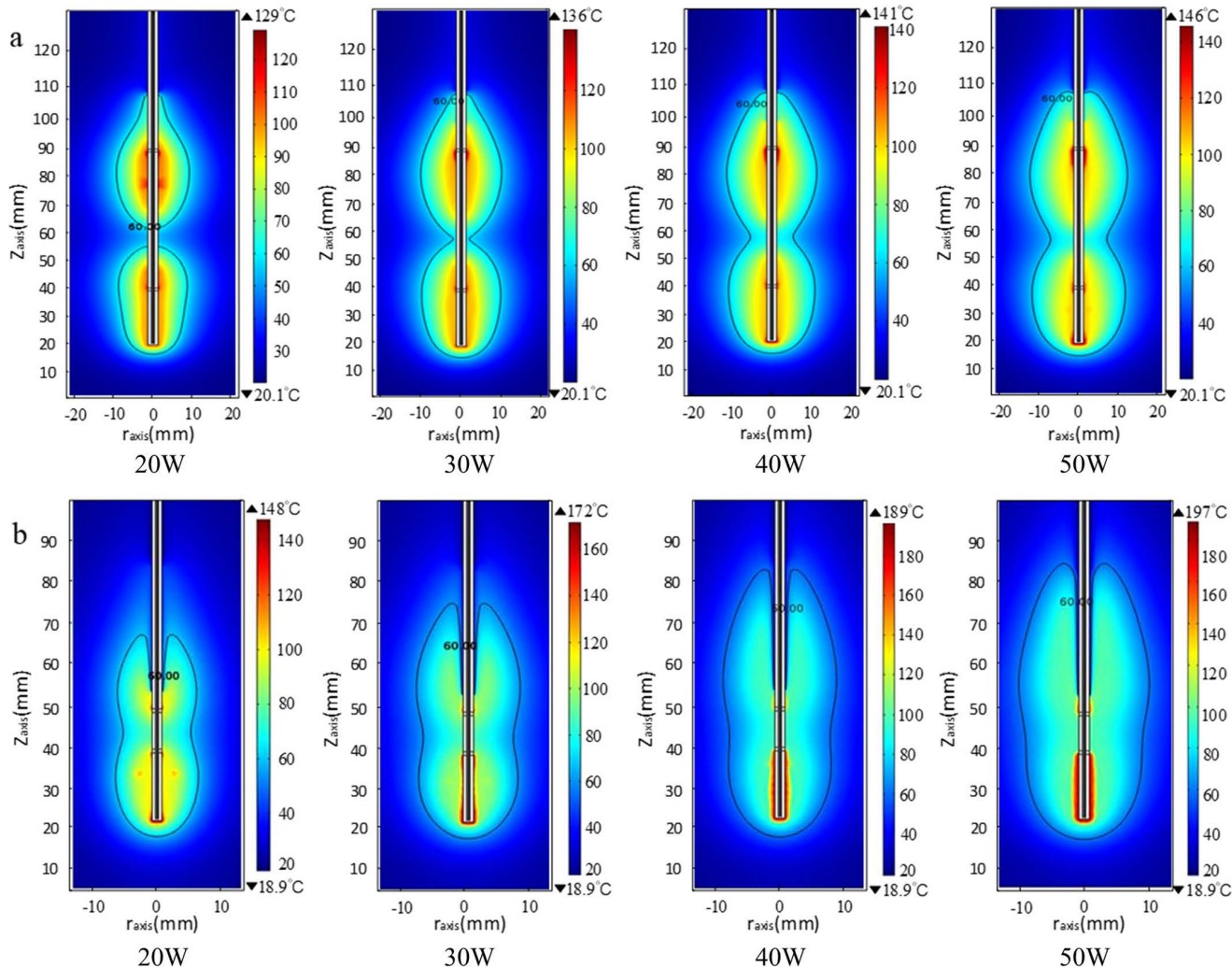

**Fig 3. Simulated temperature distributions for a heating time of 600 s based on a frequency of 433 MHz using (a) the long antenna (b) the short antenna.** From left to right, the input microwave power was set at 20, 30, 40 and 50 W, respectively.

while a 'guitar' shape formed along the short antenna. The highest values in both cases were found in the vicinity of the antenna slots and decrease with the distance. We defined the two ablation zones as ablation zone 1 (near slot 1) and 2 (near slot 2), and measured LAD and SAD for each zone area for further analysis. Based on our results, as to the long antenna, the shape and size of two ablation zones are similar (P = 0.481), and LAD is relatively larger than SAD at low microwave power (P < 0.001). The boundary of two ablation zones became less distinguishable in the case of short antenna, and the LAD and SAD of ablation zone 1 and 2 are comparable. Compared with the long antenna, higher temperature and significantly shorter LAD (P = 0.005) in the simulated ablation zone was obtained with the short antenna.

Experimentally observed ablation zones after MWA of 20, 30, 40 and 50 W for 10 min are shown in Fig 4. The observed shapes of *ex vivo* experimental ablation zones were in good agreement with patterns predicted by simulation models. Statistical analysis results also confirmed this observation, showing no significant difference of the morphologic dimensions of ablation zones between numerical simulation and experimental results. Typical 'gourd' shape ablation

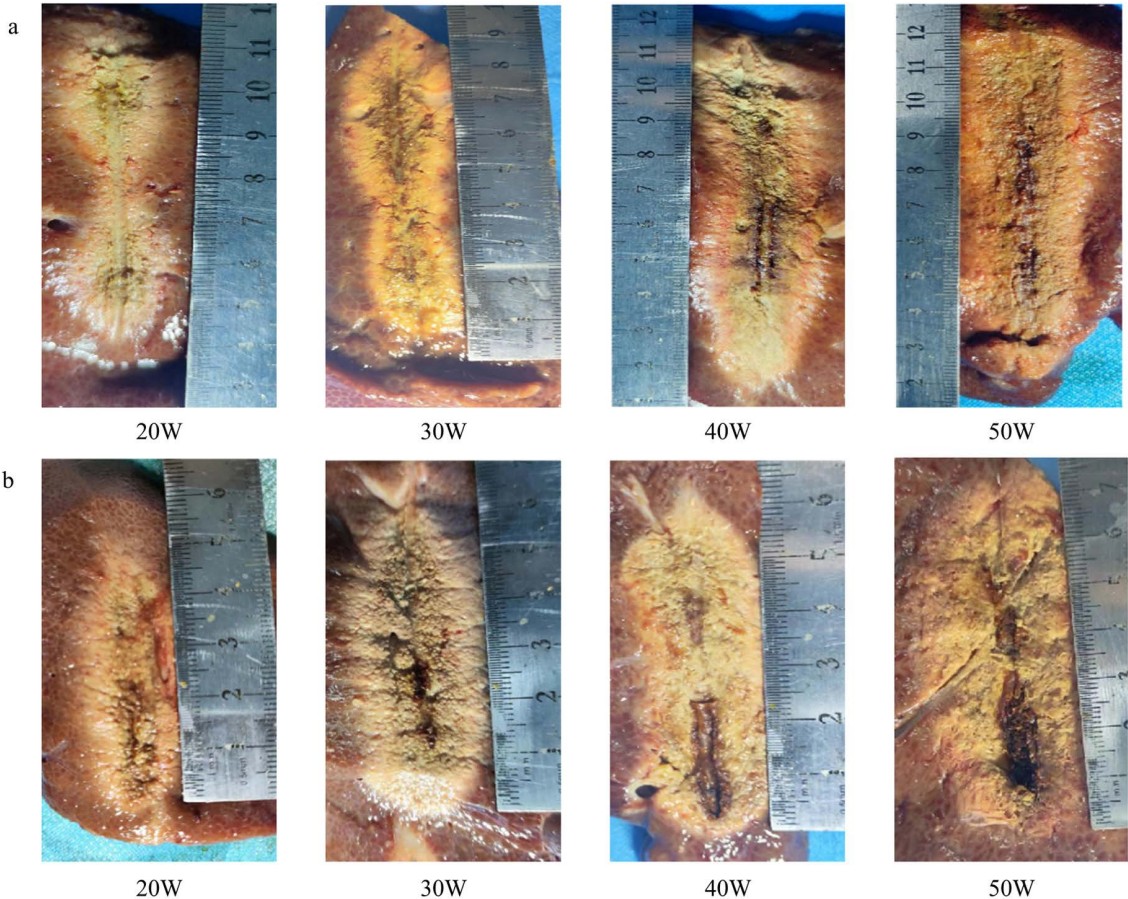

**Fig 4. Ablation of swine liver for a heating time of 600 s based on a frequency of 433 MHz using (a) the long antenna (b) the short antenna.** From left to right, the input microwave power was set at 20, 30, 40 and 50 W, respectively.

zone was noticed in the liver tissue treated with long antenna, and lower input microwave power tended to result in more distinctive ablation zones. On the other hand, the ablation zone exhibited nearly a guitar -shape after MWA using short antenna, and the boundary between the two ablation zones can hardly be determined. With a close-up observation, the ablated area adjacent the antenna slots showed carbonization features due to effect of high temperature. The morphologic characteristics of each ablated zone were assessed macroscopically with calipers, and the corresponding dimensions (LAD and SAD) were listed in Table 1.

Figs 5 and 6 summarized the effects of heating power on the size and shape of ablation zone by the use of the long and short antenna. The most significant differences of ablation results between the long and short antenna were noted for the outcomes of volume and sphericity index. Fig 5 shows that the ablation zone 1 is similar to the ablation zone 2 with the long antenna and the ablation zone 2 is slightly larger than the ablation zone 1 with the short antenna. What's more, the volume of ablation zone 1 and ablation zone 2 with the long antenna are slightly larger than that with the short antenna, respectively.

Fig 6 illustrates that although the spherical index of two ablation zones increases slightly with the increase of powers, the ablation zones formed by this double-slot microwave antennas with frequency of 433 MHz is relatively low (SI < 0.5). In comparison to Fig 6a and 6b, we can also find that the short antenna can created relatively more circular ablations compared with the long antenna.

**Table 1. The dimensions of two ablation zones with the long and short antenna in a range of heating powers for 10 min.**

| Dimensions (cm) | The long antenna | | | | The short antenna | | | |
|---|---|---|---|---|---|---|---|---|
| | 20 W | 30 W | 40 W | 40 W | 20 W | 30 W | 40 W | 50 W |
| Ablation zone 1 | | | | | | | | |
| SAD | 1.4±0.2 | 1.5±0.1 | 1.7±0.2 | 1.9±0.2 | 1.0±0.1 | 1.3±0.1 | 1.5±0.2 | 2.0±0.2 |
| LAD | 3.2±0.6 | 3.6±0.4 | 3.9±0.5 | 4.2±0.4 | 2.4±0.4 | 2.8±0.1 | 2.8±0.2 | 2.8±0.4 |
| Ablation zone 2 | | | | | | | | |
| SAD | 1.5±0.2 | 1.6±0.2 | 1.8±0.3 | 2.0±0.2 | 1.2±0.2 | 1.5±0.2 | 1.7±0.3 | 2.3±0.3 |
| LAD | 3.7±0.4 | 3.8±0.2 | 4.1±0.4 | 4.6±0.2 | 2.6±0.3 | 2.7±0.2 | 3.4±0.1 | 3.5±0.4 |

SAD: Maximum longitudinal dimension of the ablation zone along the antenna insertion path; LAD: Maximum dimension transverse to the antenna.

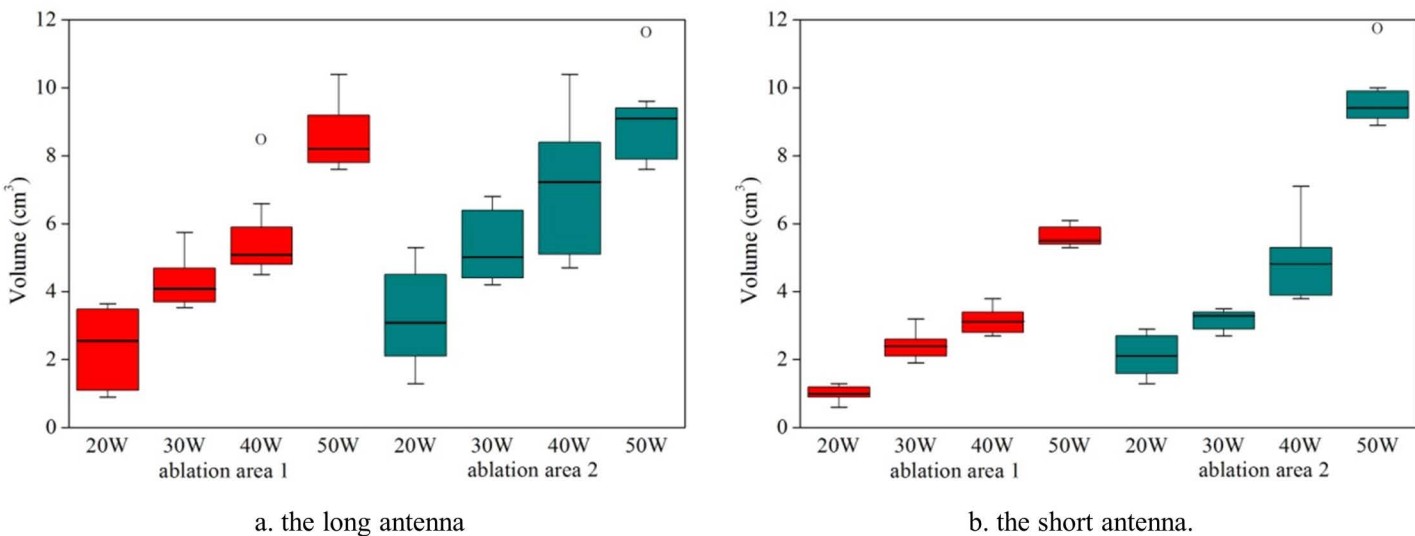

a. the long antenna                                    b. the short antenna.

**Fig 5. Box-and-whisker plot shows the effects of various heating powers on resultant damaged tissue volume after MWA using two antennas.** (a) The long antenna (b) the short antenna.

## Discussion

MWA has proven to be a successful therapeutic tool in the treatment of various types of cancer. Recent developments of MWA modalities aimed at creating larger and rounder ablation zones for hepatic cancers by sequential activation or simultaneous energy delivery coupled with multiple antennas [19–21]. Aside from hepatic tumors of which majorities are spherically shaped, there are a large amount of other types of tumors with complex structures and irregular geometry. Meanwhile, it is always crucial to avoid thermal damage to adjacent structures such as blood vessels, bowel, nerves or other organs with accurate surgical planning and precise control of thermal dose in operation [8,22,23]. As the performance and effectiveness of MWA is closely related to the antenna design, this study presents the theoretical and experimental research of *ex vivo* liver tissues using two DSMAs with delicate design.

The two types of antennas in our study can both achieve double-zone ablation, and the ablation shape can be confined to be 'gourd' and 'guitar'-like structure by tuning the length and slot-to-slot distance. Based on both theoretical and experimental results, the 'double-zone' feature was much more obvious in the case of long antenna as compared to that of

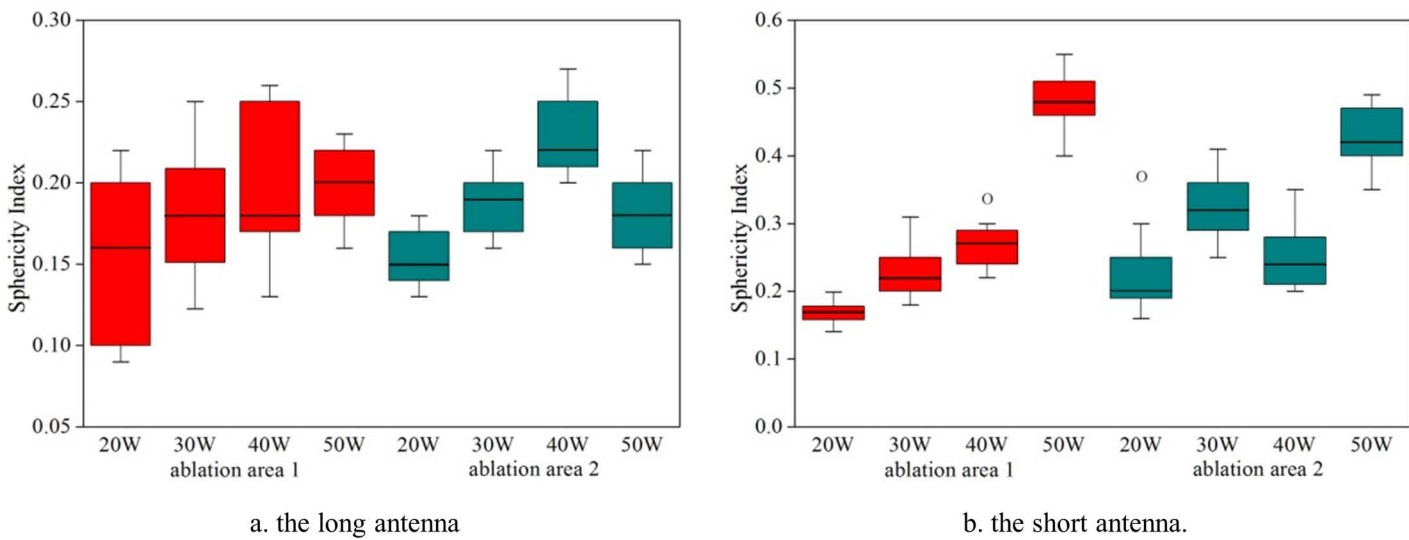

a. the long antenna b. the short antenna.

**Fig 6. Box-and-whisker plot shows the effects of various heating powers on the sphericity index of resultant damaged tissue after MWA using two antennas.** (a) The long antenna (b) the short antenna.

short antenna. This phenomenon is related to the reduction in both the length of the short antenna and the spacing between slits. As the slit spacing decreases, electromagnetic radiation at the two slits experiences significant wave interference, which markedly affects temperature distribution. Consequently, this leads to an excessive merging of the two ablation zones, resulting in unclear boundary profiles. In fact, variations in the morphology of ablation zones are closely linked to the radiation performance of antennas, with input reflection coefficient being a primary factor influencing this performance. The input reflection coefficient S11 serves as a physical quantity that reflects how well-matched a microwave antenna is with tissue impedance. Therefore, it is essential to insert a well-designed slit antenna into fresh liver tissue and measure its S11 parameters using a vector network analyzer. The results indicate that for the long needle antenna, during MWA, S11 initially decreased from -10.119 dB to -12.283 dB before gradually rising and stabilizing, and ultimately maintaining stability within a range of -9.460 dB to -9.450 dB during late-stage ablation, which essentially meeting design requirements. In contrast, for the short needle antenna, during MWA, S11 initially decreased from −15.006 dB to −17.131 dB before also experiencing an upward trend and eventual stabilization, and S11 remained stable within a range of −9.991 dB to −9.988 dB during late-stage ablation, which again largely satisfying design criteria.

Comparing the morphologic characteristics of each ablated zone, apparent charring areas were found in the livers after MWA using the short antenna, while the long antenna mainly led to tissue coagulation under the same power. The carbonization nearby the slots in the short antenna suggested that the temperature could be significantly increased with the decrease of the antenna length and slot-to-slot distance, even though circulating cooling water is provided. Intensive energy deposition induced the charring effect around the antenna tip and slot 1, resulting in smaller coagulation area observed in that area (P = 0.018; Fig 3b). Therefore, the stripe-shape morphology produced by the short antenna shows a slight difference of the size between the two ablation zones with relatively smaller area in the ablation zone 1 and became the 'guitar'-like structure with the increase of input power. In fact, the variation in the morphology of the ablation regions is closely related to the radiation

performance of antennas, with the structure at the top of the antenna being a primary factor influencing this performance. The apex of a monopole antenna consists of an inner conductor and polytetrafluoroethylene (PTFE) dielectric, whereas the tip of a slot antenna features a complete three-layer structure comprising an inner conductor, dielectric, and outer conductor. For long antennas, the ablation regions formed at both slots are more controllable; this is associated with both the longitudinal length of short antennas and reduced spacing between slots. As a result of decreased slot spacing, there is significant wave interference from electromagnetic radiation at these two slots. This interference markedly affects temperature distribution, ultimately leading to excessive merging of both ablation regions and resulting in indistinct boundary profiles.

Clearly, different design of DSMA contributed to distinctive ablation morphology. Ablated zones with longer longitudinal distance and clear boundary between the two ablations of comparable size can be achieved by the use of long antenna with long slot-to-slot distance. Whereas, the short antenna is suitable to create confluent strip-shaped ablation zones with slight size difference between the two zones. One potential application of our fabricated antenna is for heating liver cancer that are adjacent to critical structures or with stripe-shape structure. Aside from liver cancer, it could potentially allow an alternate way for treatment of colorectal cancer or bone cancer with elongated tumor morphology or multiple lesion sites. Moreover, consistent with our previous findings, effective thermal coagulation could be achieved by the use of MWA with the frequency of 433 MHz. Uniform heating characteristics and effective dose control can always be achieved with the design of our single applicator, which is also preferable for its less invasiveness. Another potential advantage of the double/ multiple slot antenna design is its relatively simple fabrication process. The specific shape and size of ablation zones can be further optimized by selecting appropriate antenna length, number and distance of the slots, and input microwave power based on the clinicians' need.

Besides the successful demonstration of two-zone ablation using double-slot MWA applicator, we also evaluated the dependence of the effectiveness of MWA on the axial length and slot-to-slot distance in a range of heating powers through both numerical and experimental studies. To our interest, simulation results agreed well with *ex vivo* results in all experimental settings, which suggested that numerical simulation could be clinically beneficial for serving as a guideline to evaluate the device performance and resultant tissue damage prior to MWA process. Indeed, some error still exists between temperature profile in tissue obtained from numerical simulation and *ex vivo* destructions. Especially, for the groups treated with lower input microwave power (20 W and 30 W), the actual ablation areas in *ex vivo* liver tissue were slightly smaller than the theoretically calculated results, which could be attributed to the possible energy loss through the transmission cable along the antenna [24]. More precise predictions of temperature distributions and tissue responses can be achieved through a more comprehensive understanding of the dynamic electric conductivity of transmission cables in future applications. Most importantly, the effects of temperature-dispersive dielectric properties and the thermal characteristics of realistic tissues should be thoroughly investigated on a case-specific basis. Furthermore, additional exploration of our antenna models must be validated for their effectiveness in vivo to support prospective clinical applications.

In summary, we designed two types of double-slot antennas for MWA with frequency of 433 MHz, which is applicable to achieve double-zone ablation with sufficient effectiveness. The delicate structure of our DSMAs affording novel ablation shape offers an effective and practical option for the clinicians to treat those tumors with elongated tumor morphology or multiple lesion sites. The dimensions of tumor heating area could be optimized by controlling the axial length, slot-to-slot distance and input power, which may pave the way for future application of patient-specific MWA in cancer treatment.

## Conclusion

Two-zone ablation can be created by a single antenna with double slots at an operating frequency of 433 MHz. The axial length and slot-to-slot distance of double-slot antenna and the input microwave power determine the temperature distributions and resultant tissue damage. Our fabricated antennas can ablate cancers that are with specific morphology or multiple lesion sites, which expands current clinical use of microwave ablation towards treating patient-specific tumors.

## Supporting information

**S1 Table. The dimensions of two ablation zones with the long and short antenna in a range of heating powers for 10 min.**
(XLS)

## Author contributions

**Conceptualization:** Zhiyu Qian.

**Data curation:** Lu Qian.

**Formal analysis:** Lu Qian, Yamin Yang.

**Methodology:** Mengwei Jiang, Weitao Li.

**Project administration:** Zhiyu Qian.

**Resources:** Zhiyu Qian.

**Software:** Mengwei Jiang.

**Supervision:** Ling Tao.

**Validation:** Weitao Li.

**Visualization:** Lidong Xing.

**Writing – original draft:** Xiaofei Jin.

**Writing – review & editing:** Ling Tao, Yamin Yang, Lidong Xing, Zhiyu Qian, Weitao Li.

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
