## [Decision Letter · Decision Letter 0]

18 Sep 2024

PONE-D-24-27447Effect of 433 MHz Double-slot Microwave Antennas for Double-zone Ablation in Ex Vivo Swine Liver ExperimentPLOS ONE

Dear Dr. Li,

Thank you for submitting your manuscript to PLOS ONE. After careful consideration, we feel that it has merit but does not fully meet PLOS ONE’s publication criteria as it currently stands. Therefore, we invite you to submit a revised version of the manuscript that addresses the points raised during the review process.

**ACADEMIC EDITOR: ** The reviewers have raised some concerns/ queries which need to be addressed point-by-point in a revised submission. To improve chances of acceptance, I recommend the following revisions: Clearly define the electromagnetic characteristics of the antennas used and explain their role in the near field, ensuring the electromagnetic principles are detailed. Additionally, consider how the electromagnetic and heat fields may influence each other during both the experiments and simulations. In the introduction, clearly justify the use of the 433 MHz frequency and provide an analysis of key antenna parameters such as reflection coefficient, impedance, and gain, and discuss how these affect the overall performance. Please also clarify how differences between experiments and simulations were quantified. As an antenna paper its important that authors compare the performance of their antennas with previous studies to contextualize their findings. Addressing these revisions will improve the clarity and impact of this manuscript.==============================

We look forward to receiving your revised manuscript.

Kind regards,

Muhammad Zubair

Academic Editor

PLOS ONE

Journal requirements: 1. When submitting your revision, we need you to address these additional requirements. Please ensure that your manuscript meets PLOS ONE's style requirements, including those for file naming. The PLOS ONE style templates can be found at https://journals.plos.org/plosone/s/file?id=wjVg/PLOSOne_formatting_sample_main_body.pdf and https://journals.plos.org/plosone/s/file?id=ba62/PLOSOne_formatting_sample_title_authors_affiliations.pdf. 2. PLOS requires an ORCID iD for the corresponding author in Editorial Manager on papers submitted after December 6th, 2016. Please ensure that you have an ORCID iD and that it is validated in Editorial Manager. To do this, go to ‘Update my Information’ (in the upper left-hand corner of the main menu), and click on the Fetch/Validate link next to the ORCID field. This will take you to the ORCID site and allow you to create a new iD or authenticate a pre-existing iD in Editorial Manager. 3. We note that the grant information you provided in the ‘Funding Information’ and ‘Financial Disclosure’ sections do not match.  When you resubmit, please ensure that you provide the correct grant numbers for the awards you received for your study in the ‘Funding Information’ section. 4. We note that your Data Availability Statement is currently as follows: [All relevant data are within the manuscript and its Supporting Information files.] Please confirm at this time whether or not your submission contains all raw data required to replicate the results of your study. Authors must share the “minimal data set” for their submission. PLOS defines the minimal data set to consist of the data required to replicate all study findings reported in the article, as well as related metadata and methods (https://journals.plos.org/plosone/s/data-availability#loc-minimal-data-set-definition). For example, authors should submit the following data: - The values behind the means, standard deviations and other measures reported;- The values used to build graphs;- The points extracted from images for analysis. Authors do not need to submit their entire data set if only a portion of the data was used in the reported study. If your submission does not contain these data, please either upload them as Supporting Information files or deposit them to a stable, public repository and provide us with the relevant URLs, DOIs, or accession numbers. For a list of recommended repositories, please see https://journals.plos.org/plosone/s/recommended-repositories. If there are ethical or legal restrictions on sharing a de-identified data set, please explain them in detail (e.g., data contain potentially sensitive information, data are owned by a third-party organization, etc.) and who has imposed them (e.g., an ethics committee). Please also provide contact information for a data access committee, ethics committee, or other institutional body to which data requests may be sent. If data are owned by a third party, please indicate how others may request data access. 5. We note that you have included the phrase “data not shown” in your manuscript. Unfortunately, this does not meet our data sharing requirements. PLOS does not permit references to inaccessible data. We require that authors provide all relevant data within the paper, Supporting Information files, or in an acceptable, public repository. Please add a citation to support this phrase or upload the data that corresponds with these findings to a stable repository (such as Figshare or Dryad) and provide and URLs, DOIs, or accession numbers that may be used to access these data. Or, if the data are not a core part of the research being presented in your study, we ask that you remove the phrase that refers to these data.

Additional Editor Comments:

Authors are requested to revise the manuscript addressing all the concerns raised by expert reviewers; A revised submission can be considered.

Reviewers' comments:

Reviewer's Responses to Questions

**Comments to the Author**

1. Is the manuscript technically sound, and do the data support the conclusions?

Reviewer #1: Yes

Reviewer #2: No

Reviewer #3: Partly

2. Has the statistical analysis been performed appropriately and rigorously? 

Reviewer #1: Yes

Reviewer #2: No

Reviewer #3: Yes

3. Have the authors made all data underlying the findings in their manuscript fully available?

Reviewer #1: Yes

Reviewer #2: Yes

Reviewer #3: Yes

4. Is the manuscript presented in an intelligible fashion and written in standard English?

Reviewer #1: Yes

Reviewer #2: Yes

Reviewer #3: Yes

5. Review Comments to the Author

Reviewer #1: The paper use two antenna to study the MWA effect, the method is simple, and it has experiment effect to surpport.

Befroe publish, there are some issue need considered.

1. How about the electromagnetic characteristics of these antennas?

2. In the paper, these antenna mainly to heat the organic tissues，which just work in the near feild, not a antenna.

3. So, the author had better give the clear electromagnetic principle of the "antenna".

4. the MWA works with electromagnetic field, heat field, is there some effect to affect these two field each other? how to consider in the simulation and experiment.

Reviewer #2: 1. The main controbution is not clear, Authors should add the main contributions of the proposed work in the introduction.

2. This paper proposes Double-slot Microwave Antennas for Double-zone Ablation in Ex Vivo Swine Liver Experiment. However, the author has not explained how the steps to design and design the antenna? The author needs to add the steps in designing the antenna.

3. The author needs to explain the reason for using the low frequency of 433 MHz?

4. Generally, the observed antenna parameters are reflection coefficient, impedance and gain. In this paper, the observed effects of the antenna are not clear, how does it affect the resonance frequency? reflection coefficient and impedance?

Reviewer #3: This study provides a foundation for the development of advanced MWA techniques that can be tailored to individual patient needs, potentially improving outcomes for patients with complex tumor geometries. (major revision)

1. The objective is clearly stated in the 'Purpose' section of the abstract; however, the conclusion could more effectively emphasize the practical applicability of this research. For example, it could better connect the results to potential clinical applications beyond cancer therapies.

2. The terms describing ablation zones such as “gourd shape” or “guitar shape“, should be frequent and used consistently throughout the manuscript. And if this terminology does only first appear in the results, then a short description should be delivered already at an earlier stage of the Methods chapter.

3. The details of the ex vivo experiment might be more elegant. Essential experimental controls and the ways by which they were maintained should have been better detailed, especially on how temperature control of liver samples was achieved before ablation.

4. It should be justified on why the particular type of statistical tests were performed and how to interpret them. What, for example, was the specific quantification of those differences between experiment and simulation?

5. How antenna performance effect on whole experiment and please provide previous study antenna comparison according to the experiment.

6. PLOS authors have the option to publish the peer review history of their article (what does this mean? ). If published, this will include your full peer review and any attached files.

**Do you want your identity to be public for this peer review?** For information about this choice, including consent withdrawal, please see our Privacy Policy .

Reviewer #1: No

Reviewer #2: No

Reviewer #3: No

---

## [Author Response · Author response to Decision Letter 1]

28 Oct 2024

Dear Prof. Muhammad Zubair:

Thank you for your letter and for the reviewers’ comments concerning our manuscript entitled “Effect of 433 MHz Double-slot Microwave Antennas for Double-zone Ablation in Ex Vivo Swine Liver Experiment” (ID: PONE-D-24-27447). These advices are all valuable and very helpful for revising and improving our paper, as well as the important guiding significance to our studies. We have made corresponding revision according to these advices. Words in blue are the changes I have made in the text. The following is the answers and revisions I have made in response to these questions and suggestions item by item.

Responses to the reviewer’s comments:

Reviewer 1:

Responses to comments:

1. How about the electromagnetic characteristics of these antennas?

Reply: Thank you very much for the valuable comment. In the electromagnetic characteristics of the microwave ablation needle, the S11 parameter serves as an indicator of microwave energy reflection at a specific frequency. This parameter is crucial for evaluating the performance of the ablation needle. We have provided additional information in our newly submitted manuscript.

The main modifications are as follows:

The two types of antennas in our study can both achieve double-zone ablation, and the ablation shape can be confined to be ‘gourd’ and ‘guitar’-like structure by tuning the length and slot-to-slot distance. Based on both theoretical and experimental results, the ‘double-zone’ feature was much more obvious in the case of long antenna as compared to that of short antenna. This phenomenon is related to the reduction in both the length of the short antenna and the spacing between slits. As the slit spacing decreases, electromagnetic radiation at the two slits experiences significant wave interference, which markedly affects temperature distribution. Consequently, this leads to an excessive merging of the two ablation zones, resulting in unclear boundary profiles. In fact, variations in the morphology of ablation zones are closely linked to the radiation performance of antennas, with input reflection coefficient being a primary factor influencing this performance. The input reflection coefficient S11 serves as a physical quantity that reflects how well-matched a microwave antenna is with tissue impedance. Therefore, it is essential to insert a well-designed slit antenna into fresh liver tissue and measure its S11 parameters using a vector network analyzer. The results indicate that for the long needle antenna, during MWA, S11 initially decreased from -10.119 dB to -12.283 dB before gradually rising and stabilizing, and ultimately maintaining stability within a range of -9.460 dB to -9.450 dB during late-stage ablation, which essentially meeting design requirements. In contrast, for the short needle antenna, during MWA, S11 initially decreased from -15.006 dB to -17.131 dB before also experiencing an upward trend and eventual stabilization, and S11 remained stable within a range of -9.991 dB to -9.988 dB during late-stage ablation, which again largely satisfying design criteria.

2. In the paper, these antenna mainly to heat the organic tissues, which just work in the near feild, not a antenna

Reply: Thank you very much for the valuable comment. We have carefully revised the manuscript according to suggestions, supplemented in the discussion section, and the revised parts are in blue fonts.

The main revisions are listed as follows:

In fact, the variation in the morphology of the ablation regions is closely related to the radiation performance of antennas, with the structure at the top of the antenna being a primary factor influencing this performance. The apex of a monopole antenna consists of an inner conductor and polytetrafluoroethylene (PTFE) dielectric, whereas the tip of a slot antenna features a complete three-layer structure comprising an inner conductor, dielectric, and outer conductor. For long antennas, the ablation regions formed at both slots are more controllable; this is associated with both the longitudinal length of short antennas and reduced spacing between slots. As a result of decreased slot spacing, there is significant wave interference from electromagnetic radiation at these two slots. This interference markedly affects temperature distribution, ultimately leading to excessive merging of both ablation regions and resulting in indistinct boundary profiles.

3. So, the author had better give the clear electromagnetic principle of the "antenna".

Reply: Thank you very much for the valuable comment. These related works have been included in introduction. In the third paragraph of the Introduction, “The performance of an antenna, particularly its capability for backward heating along the axial direction, is significantly influenced by its structural design. Slit antennas demonstrate higher radiation efficiency and reduced energy loss compared to monopole and dipole antennas, rendering them ideal for microwave ablation needle applications. ” have been added.

In addition, an electromagnetic model of the antenna was added to the manuscript.

The main revisions are listed as follows:

In the electromagnetic field distribution of a coaxial microwave antenna, both the electric vector and magnetic vector are perpendicular to the direction of propagation; specifically, the electric field has values only in the radial (r) component, while the magnetic field is present solely along the azimuth (φ) axis. Therefore, this paper employs a transverse magnetic (TM) wave equation as the model for electromagnetic wave propagation, represented by the following equation:

3

Where, indicates the relative permittivity in a vacuum ,which has the value of 8.854×10−12 F/m）, , and are the relative permittivity, electrical conductivity and permeability of liver tissue respectively, denotes the free-space wave number, is the magnetic field strength

The Specific Absorption Rate (SAR) reflects the microwave energy absorbed per unit of time by biological tissues and can be expressed as:

4

Where, denotes the induced electric field in the organization, is the liver tissue density.

4. the MWA works with electromagnetic field, heat field, is there some effect to affect these two field each other? how to consider in the simulation and experiment.

Reply: We sincerely thank the reviewer for careful reading. The simulation of the temperature field in the microwave ablation dose control model primarily relies on the interdependent and coupled electromagnetic wave and bioheat conduction models. We have taken care to include additional details regarding the interaction between these electromagnetic and bio-thermal fields in the Numerical Simulation Study section of Materials and Methods. We have carefully revised the manuscript according to suggestions and the revised parts are in blue fonts.

The main revisions are listed as follows:

The numerical simulation models of swine liver and antennas were set up in COMSOL Multiphysics (COMSOL 5.1, Stockholm, Sweden) using Finite Element Method (FEM). Since the isolated pig liver serves as a more homogeneous medium, we assume that the liver tissue is isotropic. This allows us to simplify the simulation model of the liver tissue, which originally possesses an irregular shape, into a two-dimensional axially symmetric model. In the ablation experiment, the depth of insertion for the 433 MHz microwave ablation needle into the isolated liver tissue will not exceed 14 cm. Consequently, we set the height of the three-dimensional cylinder to 140 mm and its transverse diameter to 50 mm. The depth of antenna insertion into the liver tissue will be twice that of its axial length; thus, for long antennas it will measure 140 mm and for short antennas it will measure 60 mm.

The simulation of the temperature field in the microwave ablation model primarily utilizes a coupled and interdependent electromagnetic wave and bioheat transfer model. In the electromagnetic field distribution of a coaxial microwave antenna, both the electric vector and magnetic vector are perpendicular to the direction of propagation; specifically, the electric field has values only in the radial (r) component, while the magnetic field is present solely along the azimuth (φ) axis. Therefore, this paper employs a transverse magnetic (TM) wave equation as the model for electromagnetic wave propagation, represented by the following equation:

3

Where, indicates the relative permittivity in a vacuum ,which has the value of 8.854×10−12 F/m）, , and are the relative permittivity, electrical conductivity and permeability of liver tissue respectively, denotes the free-space wave number, is the magnetic field strength

The Specific Absorption Rate (SAR) reflects the microwave energy absorbed per unit of time by biological tissues and can be expressed as:

4

Where, denotes the induced electric field in the organization, is the liver tissue density.

After biological tissues absorb microwave energy, the energy is subsequently transmitted within the tissue in the form of heat. This transmission primarily occurs through two mechanisms: thermal conduction and convection. In the field of clinical microwave hyperthermia, accurately predicting and controlling the temperature distribution in tissues following energy absorption is essential. This necessitates the development of a model for heat conduction in biological tissues. Among various models describing temperature field variations, Pennes' bioheat equation stands out as the most classical representation, expressed as follows:

5

where , , and represent the density, specific heat capacity, and thermal conductivity of liver tissue, respectively. In the right-hand term of the equation, accounts for heat transfer between tissues, while denotes the blood flow action term. The variables , , , correspond to the density, specific heat capacity, perfusion rate, thermal conductivity, and temperature of blood. Additionally, terms and are defined as representing metabolic heat production within the tissue and as denoting heat load applied by an external source (specifically from microwave electromagnetic fields), respectively. The external heat source term is associated with Specific Absorption Rate (SAR) and can be further expressed as follows:

6

According to Eqs. 6, the temperature field during microwave ablation can be simulated by solving both the planar transverse magnetic field wave equation and the Pennes' bioheat equation.

In numerical simulations, the thermal and dielectric properties of the liver tissue model are critical biophysical parameters that vary with tissue temperature and water content throughout the heat transfer process in biological tissues. This paper presents expressions below that incorporate appropriate corrections based on previous literature:

7

8

9

10

11

The insulating medium of the microwave ablation needle body was polytetrafluoroethylene (PTFE), characterized by a dielectric constant of 2.03 and zero conductivity. A boundary temperature of 15 °C was applied to the outer conductor surface of the ablation needle model to simulate the effects of circulating water cooling. The simulation model represented ex vivo hepatic tissue, with both blood perfusion rate and metabolic heat production set to zero; additionally, the initial temperature of the hepatic tissue was established at 15 °C.

Reviewer 2:

Responses to comments:

1. The main controbution is not clear, Authors should add the main contributions of the proposed work in the introduction.

Reply: Thank you very much for the valuable comment. We have carefully revised the manuscript according to suggestions and the revised parts are in blue fonts.

The main revisions are listed as follows:

In present study, we successfully designed two coaxial slit antennas based on the 433 MHz microwave frequency, so that MWA can be used to treat tumors of a specific shape, size, and location. The design of our double slot microwave antennas (DSMAs) with axial lengths of 70 mm and 30 mm were optimized by numerical simulation and ex vivo liver experiments. Different axial length and slot-to-slot distance of DSMAs were also studied to evaluate the effectiveness of MWA in a range of heating powers. The dedicated structure of our DSMAs with frequency of 433 MHz could create novel ablation shapes, including gourd-shape and guitar-shape ablation zones with controllable dimensions, which owns potential for future application of precise and patient-specific MWA.

2. This paper proposes Double-slot Microwave Antennas for Double-zone Ablation in Ex Vivo Swine Liver Experiment. However, the author has not explained how the steps to design and design the antenna? The author needs to add the steps in designing the antenna.

Reply: Thank you very much for the valuable comment. We have carefully revised the manuscript according to suggestions and the revised parts are in blue fonts.

The main revisions are listed as follows:

Antenna Design

In this study, two coaxial slit microwave antennas have been designed, both of which use semi-rigid coaxial cable (SFT-50-1) with a characteristic impedance of 50 Ω. The specifications of the cable include an inner conductor diameter of 0.3 mm, an insulating dielectric diameter of 0.8 mm, and an outer conductor diameter of 1.18 mm. At the top of each antenna, the inner and outer conductors are welded together to form a conductive cap. By removing the rings from the outer conductor of the coaxial cable slots are formed, which are channels for the electromagnetic wave radiation energy to the biological tissues. The theoretical value of the slit width ( ) is related to the effective wavelength ( ) and the relative dielectric constant ( ) and is calculated as follows:

1

The theoretical value of the gap spacing ( ) is also related to the effective wavelength ( ), which needs to meet the requirement of equal amplitude and same phase feeding, and the general calculation formula is shown below:

2

The dimensions of the antenna structures are optimised on the basis of the effective wavelength ( ) taking into account the results of theoretical calculations and previous experience [5]. The DSMA with axial length of 70 mm (about at 433 MHz) was defined as long antenna, whereas the DSMA with axial length of 30 mm (about at 433 MHz) was defined as short antenna. The slots with the width of 1 mm in each antenna were created by removing annular portions of the outer conductor of the coaxial cable [13]. Different tip-to-slot distance and slot-to-slot distance were investigated in corresponding antenna, respectively, and the specific structural details are illustrated in Fig. 1.

The ablation needle features a conical front end and a cylindrical rear end, with a PTFE dielectric sleeve firmly attached to the slit, ensuring a seamless connection to the 304 stainless steel needle tube. Semi-rigid coaxial cable is passed through the needle bar and connected to the RF connector for connection and propagation of the 433 MHz microwave source. The water-cooled section is connected to the inlet and outlet pipes via equal-diameter plastic four-way fittings to form a simple tank structure. Cooling water is injected through a capillary tube and flows through the working part of the needle body and then out to achieve circulating water cooling. This design effectively mitigates the trailing phenomenon and tissue carbonization during the ablation process, thereby optimizing the outcomes of ablation.

3. The author needs to explain the reason for using the low frequency of 433 MHz?

Reply: Thank you very much for the valuable comment. These related works have been included in introduction. In the fifth paragraph of the Introduction, “According to microwave theory, a lower microwave frequency is associated with a more significant increase in temperature, which facilitates the heating of deeper tissues. Our previous study showed that [5], compared with antenna with frequency of 2450 MHz, larger ablation area was indeed obtained by our coaxial slot antenna with frequency of 433 MHz. This finding theoretically underscores the efficacy of microwave

---

## [Decision Letter · Decision Letter 1]

29 Nov 2024

Effect of 433 MHz Double-slot Microwave Antennas for Double-zone Ablation in Ex Vivo Swine Liver Experiment

PONE-D-24-27447R1

Dear Dr. Li,

We’re pleased to inform you that your manuscript has been judged scientifically suitable for publication and will be formally accepted for publication once it meets all outstanding technical requirements.

Kind regards,

Muhammad Zubair

Academic Editor

PLOS ONE

Additional Editor Comments (optional):

The revised manuscript has addressed the concerns raised by reviewers and is ready for acceptance.

Reviewers' comments:

Reviewer's Responses to Questions

**Comments to the Author**

1. If the authors have adequately addressed your comments raised in a previous round of review and you feel that this manuscript is now acceptable for publication, you may indicate that here to bypass the “Comments to the Author” section, enter your conflict of interest statement in the “Confidential to Editor” section, and submit your "Accept" recommendation.

Reviewer #1: All comments have been addressed

Reviewer #3: All comments have been addressed

2. Is the manuscript technically sound, and do the data support the conclusions?

Reviewer #1: Yes

Reviewer #3: (No Response)

3. Has the statistical analysis been performed appropriately and rigorously? 

Reviewer #1: Yes

Reviewer #3: Yes

4. Have the authors made all data underlying the findings in their manuscript fully available?

Reviewer #1: Yes

Reviewer #3: Yes

5. Is the manuscript presented in an intelligible fashion and written in standard English?

Reviewer #1: Yes

Reviewer #3: Yes

6. Review Comments to the Author

Reviewer #1: The authors have answered my questions one by one, and revised the manuscript according to the comments.

Reviewer #3: Comments addressed properly

7. PLOS authors have the option to publish the peer review history of their article (what does this mean? ). If published, this will include your full peer review and any attached files.

**Do you want your identity to be public for this peer review?** For information about this choice, including consent withdrawal, please see our Privacy Policy .

Reviewer #1: No

Reviewer #3: No

---

## [Editor Report · Acceptance letter]

PONE-D-24-27447R1

PLOS ONE

Dear Dr. Li,

I'm pleased to inform you that your manuscript has been deemed suitable for publication in PLOS ONE. Congratulations! Your manuscript is now being handed over to our production team.

Kind regards,

on behalf of

Dr. Muhammad Zubair

Academic Editor

PLOS ONE